# Esophagogastroduodenal Findings in Patients with Intraductal Papillary Mucinous Neoplasms

**DOI:** 10.3390/diagnostics13122127

**Published:** 2023-06-20

**Authors:** Dana Zelnik Yovel, Erwin Santo, Majd Khader, Roie Tzadok, Nir Bar, Asaf Aizic, Oren Shibolet, Dana Ben-Ami Shor

**Affiliations:** 1The Kamila Gonczarowski Institute of Gastroenterology and Liver Diseases, Shamir (Assaf Harofeh) Medical Center, Zerifin 703000, Israel; danazelnik@gmail.com; 2Sackler Faculty of Medicine, Tel Aviv University, Tel Aviv 6423906, Israel; 3Department of Gastroenterology, Tel Aviv Sourasky Medical Center, Tel Aviv 6423906, Israel; 4Department of Pathology, Tel Aviv Sourasky Medical Center, Tel Aviv 6423906, Israel

**Keywords:** intraductal papillary mucinous neoplasms (IPMNs), gastric cancer, esophageal cancer

## Abstract

The association between intraductal papillary mucinous neoplasms (IPMNs) and extra-pancreatic malignancies is controversial. This cross-sectional study compared esophagogastroduodenal findings in 340 IPMN patients to those of age- and gender-matched controls without known IPMNs who underwent esophagogastroduodenoscopies (EGDs) for similar clinical reasons. The presence of gastric and esophageal cancer, Barrett’s esophagus, neuroendocrine tumors (NETs), gastrointestinal stromal tumors (GISTs), gastric adenomas, and ampullary tumors was assessed. The results showed that 4/340 (1.2%) of the IPMN patients had gastric cancer and 1/340 (0.3%) had esophageal cancer. The matched control group had a similar incidence of gastric cancer (5/340) (1.5%), with no esophageal cancer cases (*p* > 0.999). The overall incidence of other esophagogastroduodenal conditions did not significantly differ between the IPMN patients and the controls. However, the incidence of gastric cancer in the IPMN patients was higher than expected based on national cancer registry data (standardized incidence ratio of 31.39; *p* < 0.001; CI 8.38–78.76). In conclusion, IPMN patients have a significantly higher incidence of gastric cancer compared to the general population. However, the incidence of esophagogastroduodenal findings, including gastric and esophageal cancer, is similar between IPMN patients and those who undergo an EGD for similar clinical indications. Further research is needed to determine optimal surveillance strategies for IPMN patients regarding their risk of developing gastric cancer.

## 1. Introduction

Cystic lesions of the pancreas are a common incidental finding in abdominal imaging. In recent years, their prevalence among asymptomatic patients has been on the rise, mainly due to improvements in imaging techniques. Studies have estimated that cystic lesions of the pancreas are present in approximately 2.4% to 13.5% of asymptomatic populations [1]. It is important to note that the most common pancreatic cysts encountered are intraductal papillary mucinous neoplasms (IPMNs) [1]. IPMNs are characterized by the intraductal growth of mucin-producing cells, and they can be classified into main duct, side duct (branch duct), or mixed types based on their location within the pancreatic ductal system.

IPMNs have gained significant attention in clinical practice due to their potential for malignant transformation. Studies have shown that the risk of malignancy is higher in IPMNs compared to other types of pancreatic cysts [2,3]. The incidence of IPMNs increases with age, and they are more commonly detected in individuals over 60 years of age [1].

The association between intraductal papillary mucinous neoplasms (IPMNs) and extra-pancreatic malignancies (EPMs) has been a subject of significant research interest. Several studies have reported a higher prevalence of EPMs in patients with IPMNs compared to the general population. The incidence of EPMs among patients with IPMNs has been reported to range from 10% to 52% in various studies [4,5,6,7,8,9,10,11,12,13,14,15].

A recent meta-analysis, which included a total of 8240 patients from sixteen studies, found an odds ratio of 57.9 for any EPM in the presence of an IPMN [16]. The most common types of EPMs in patients with IPMNs are colon and gastric cancers. In fact, the odds ratio for upper or lower gastrointestinal cancers (such as esophagus, stomach, colon, and rectum) in the presence of an incidental IPMN was found to be 12.9 [16].

It is worth noting that the prevalence of EPMs in patients with IPMNs reflects the geographic distribution of tumors in each country. For example, gastric cancer is the most common EPM in eastern countries, while colorectal adenocarcinoma is more prevalent in western countries [17].

The exact shared carcinogenic mechanisms between gastric cancer and intraductal papillary mucinous neoplasms (IPMNs) are not yet fully understood. However, studies have suggested some potential links and common molecular pathways between these two conditions. For example, Yamanoi et al. [18] reported that diminished αGlcNAc expression on MUC6 in precancerous lesions in the stomach (gastric gland mucin), pancreas (IPMNs), and esophagus (Barrett’s esophagus) is an early event indicative of tumor progression.

The main aim of this study was to compare the incidence rate of gastric and esophageal cancers in patients with and without a known diagnosis of IPMNs, and to the general population in Israel.

## 2. Materials and Methods

### 2.1. Study Population

The study population included 340 patients aged 18 and over who had undergone an endoscopic ultrasound (EUS) at Tel Aviv Sourasky Medical Center between 2004 and 2021 and had received a diagnosis of IPMNs. These patients also had at least one esophagogastroduodenoscopy (EGD) at the same medical center during the same period. Pregnant women, patients under 18 years of age, and those with any additional gastrointestinal diseases were excluded from the study.

For each patient, the location, size, and presence of worrisome features and high-risk stigmata of the IPMNs were assessed. The worrisome features included the following: the presence of a mural nodule, thickened cyst walls, main pancreatic duct dilation, and lymphadenopathy [19].

A control group was established that consisted of patients who had undergone an EGD for similar indications (e.g., dyspepsia, dysphagia, hematemesis/coffee ground, heartburn, melena, nausea/vomiting, weight loss, diarrhea, anemia, occult blood, suspected celiac disease, etc.) but did not have a known diagnosis of IPMNs. The control group was matched to the IPMN group based on age, gender, and EGD indications.

For each patient, only the first EGD during the study period was included. Various parameters were evaluated during the study, including the presence of new gastric and esophageal cancers, Barrett’s esophagus, gastric and duodenal neuroendocrine tumors (NETs), gastrointestinal stromal tumors (GISTs), gastric adenomas, and ampullary tumors.

Furthermore, the incidence of new gastric cancer among IPMN patients was compared to the incidence in the general population in Israel based on data from the Israel National Cancer Registry (INCR) updated until 2018.

All relevant data, including IPMN characteristics and the presence of extra-pancreatic malignancies, were retrieved from the computerized data records of Tel Aviv Sourasky Medical Center. The study was approved by the local institutional ethics committee of Tel Aviv Sourasky Medical Center.

### 2.2. Statistical Analysis

The categorical variables were presented as frequencies and percentages. Age, age at EGD, and cyst size were reported as median and interquartile range (IQR) The two cohorts were carefully matched based on age, with a maximum difference of ±1 year, and gender to ensure comparability. The statistical analysis involved the use of appropriate tests. The McNemar test was employed to compare the categorical variables between the matched groups, while the Wilcoxon test was utilized for the comparison of the continuous and ordinal variables. To assess the incidence of gastric and esophageal cancers, the standardized incidence ratio was calculated, comparing the observed incidence in the study population to the expected incidence based on age- and gender-specific proportions reported by the INCR data- updated to 2018. All statistical tests were two-sided, and a significance level of *p* < 0.05 was considered statistically significant. The statistical analysis was performed using SPSS software (IBM SPSS Statistics for Windows version 25; IBM Corp., Armonk, NY, USA, 2017).

## 3. Results

The median age of the study population was 70 years old (interquartile range [IQR]: 61–76.5) at the time of the endoscopic ultrasounds (EUSs) and 71 years old (IQR: 63–77) at the time of the esophagogastroduodenoscopies (EGDs), as shown in Table 1. Among the participants, 36.1% were male. Of the 340 patients diagnosed with an IPMN, the majority (314/340, 92.3%) had a branch-duct IPMN (SB-IPMN), 15/340 (4.4%) had a mixed-type IPMN, and 11/310 (3.2%) had a main-duct IPMN (MD-IPMN). Worrisome features, including mural nodule, cyst wall thickening, intracystic mass, lymphadenopathy, or main pancreatic duct (MPD) dilatation, were observed in 41 patients (12%) [19]. Additionally, ten patients (2.9%) in the cohort exhibited high-risk stigmata, such as an enhancing mural nodule > 5 mm, MPD > 10 mm, or obstructive jaundice [19]. The cysts’ fluid cytology and the clinical decision data are also presented in Table 1.

Comparing patients with pancreatic cysts to age- and gender-matched controls, the incidence of gastric cancer was similar between the two groups—4 out of 340 (1.2%) in the IPMN group and 5 out of 340 (1.5%) in the control group (*p* > 0.999). Likewise, the incidence of esophageal cancer was comparable in the matched individuals, with 1 out of 340 (0.3%) in the IPMN group and none in the control group (*p* > 0.999). Furthermore, the overall incidences of Barrett’s esophagus, gastric and duodenal neuroendocrine tumors (NETs), gastrointestinal stromal tumors (GISTs), gastric adenomas, and ampullary tumors did not show significant differences between the patients with IPMNs and the matched control group (Table 2).

Table 3 provides a comprehensive description of all nine cases of gastric cancer identified in our cohort, including both patients with known IPMNs and those in the control group. In the IPMN group, four cases of gastric cancer were diagnosed during the initial EGDs. These cases presented with various characteristics, such as a large ulcerated infiltrating mass at the gastric body (Figure 1a), an infiltrating mass with contact oozing at the gastric body (Figure 1b), a partially obstructing, circumferential, ulcerated, irregular mass at the gastric antrum (Figure 1c), and a large friable ulcerated mass at the gastric fundus (Figure 1d).

In a secondary analysis, the incidence of gastric cancer was found to be higher than expected in patients with IPMNs compared to the general population, as indicated by the Israel National Cancer Registry (INCR) data (standardized incidence ratio of 31.39; *p* < 0.001; CI 8.38–78.76) (Table 4).

## 4. Discussion

In a recent study [20], we assessed the prevalence of polyps and CRC in patients with IPMNs compared to an average-risk and age- and gender-matched population. We have compared colonoscopy findings in IPMN patients to an average-risk population of individuals that were age- and sex-matched. Among the patients with IPMN cysts, the prevalence of CRC was significantly higher compared to the age- and gender-paired controls—16/310 (5.2%) versus 4/310 (1.3%), respectively [*p* = 0.012; a prevalence odds ratio (POR) of 4; a confidence interval (CI) of 1.29–16.44]. Our findings supported previous studies observing an increased incidence of CRC in patients with IPMNs (8) (9) (13). The overall prevalence of colorectal polyps in our study was similar for patients with IPMNs compared to the paired group—96/310 (31%) versus 83/310 (26.8%), respectively (*p* = 0.291; POR 1.22; CI 0.85–1.76). However, the prevalence of advanced histological polyps classified as HGD adenomas was significantly higher in patients with IPMN-type pancreatic cysts when compared to the paired group—13/310 (4.2%) versus 3/310 (1%), respectively (*p* = 0.021; POR 4.33; CI 1.19–23.7). In addition, the prevalence of large polyps (20 mm or more) was found to be significantly higher in patients with IPMNs compared to the paired group—19/310 (6.1%) versus 6/310 (1.9%), respectively (*p* = 0.011; POR 3.6; CI 1.29–12.4). Interestingly, there was no significant difference in the prevalence of polyps between IPMN patients who had worrisome features and/or high-risk stigmata compared to the paired group—14/40 (31.8%) versus 14/40 (31.8%), respectively (*p* > 0.999). In addition, no significant difference was found in the presence of CRC between these groups—4/40 (9.1%) versus 1/40 (2.3%), respectively (*p* = 0.375).

In this current study, we aimed to investigate the prevalence of esophagogastroduodenal findings in patients with pancreatic IPMNs compared to age- and gender-matched controls without a previous IPMN diagnosis. The incidence of esophagogastroduodenal findings was found to be similar between patients with IPMNs and those without IPMNs who underwent endoscopic examination for similar clinical indications. However, patients with IPMNs displayed a significantly higher incidence of gastric cancer compared to the average-risk population in Israel, as indicated by data from the Israel National Cancer Registry (INCR).

Previous studies conducted in southeast Asia have consistently demonstrated an elevated risk of gastric cancer among patients with IPMNs [5,8,11,21]. However, it is important to note that these studies were primarily conducted in regions where gastric carcinomas are more prevalent compared to western countries. This geographical variation in gastric cancer incidence raises the question of whether the observed increased risk in IPMN patients can be generalized to populations in other parts of the world.

Furthermore, a potential limitation of these previous studies is their focus on cohorts consisting predominantly of patients who had undergone surgical resection for IPMNs. While these cohorts provide valuable insights into the management and outcomes of surgically treated IPMNs, they may introduce selection bias by primarily including patients with high-risk IPMNs. This selection bias could potentially overestimate the true risk of gastric cancer in the broader IPMN population, limiting the generalizability of the findings to the general IPMN patient population. Therefore, our current study contributes to the existing literature by assessing the risk of gastric cancer in a broader IPMN population, irrespective of surgical intervention. By encompassing a wider range of IPMN patients, our study aims to provide a more representative assessment of the risk of gastric cancer in IPMN populations, thus enhancing our understanding of the disease and informing clinical decision-making.

In this study, we conducted a comparison between IPMN patients and matched controls without known IPMNs who presented with similar upper gastrointestinal (GI) complaints. Surprisingly, we did not observe a significant difference in the incidence of upper GI malignancies between these two groups. However, when we utilized the Israeli National Cancer Registry to assess the prevalence of gastric cancer in the general Israeli population, we found that IPMN patients had an increased risk of developing this type of cancer. This finding suggests a potential association between IPMNs and gastric neoplasms. This difference may be attributed to the initial reasons for performing esophagogastroduodenoscopies (EGDs) in the control group, indicating a correlation between upper GI complaints and the development of gastric neoplasms.

Utilizing a nationwide registry to compare the observed incidences of extra-pancreatic malignancies (EPMs) among IPMN patients to the expected incidence is an effective approach. However, a comprehensive review of the literature revealed that data in this area are limited. For example, a study conducted by Kawakubo et al. [22] prospectively followed 642 IPMN patients for an average of 4.8 years. The incidence of observed extra-pancreatic malignancies, including gastric cancer, in this study was similar to the expected incidence in an age- and gender-matched general Japanese population based on the vital statistics of Japan. In contrast, another study by Eguchi et al. [8] found a significantly higher overall incidence of EPMs among IPMN patients compared to the general population, primarily driven by a significant increase in colorectal cancer. However, gastric cancer did not exhibit a significant increase in this study. It is important to note that both of these studies were based on cohorts from Japan, which may have distinct characteristics and risk factors compared to other populations.

In the current study, we evaluated the presence of worrisome features and high-risk stigmata, which are recognized as risk factors for the malignant development of cysts, and upper endoscopy findings in relation to these factors. However, we did not find a significant association between the presence of worrisome features/high-risk stigmata and the upper endoscopy findings in IPMN patients. It is important to note that the number of IPMN cysts exhibiting worrisome features or high-risk stigmata was relatively small, which may have limited our ability to reach statistical significance in this regard.

The understanding of a potential shared pathogenesis between esophageal/gastric tumors and pancreatic cancer is still limited, and further research is needed to elucidate the underlying mechanisms. However, some studies have provided insights into potential molecular factors and genetic alterations that may contribute to the development of extra-pancreatic gastrointestinal cancers. For example, Lee et al. [23] reported a potential relationship between the transcription of the MUC2 gene and the development of synchronous extra-pancreatic gastrointestinal cancers in patients with IPMNs. Additionally, GNAS and KRAS mutations have been frequently observed in pyloric gland adenoma, IPMNs, and pancreatic intraepithelial neoplasia, respectively [24,25,26]. These mutations indicate aberrant signaling pathways that contribute to the initiation and progression of these neoplasms. In a review article, Yamanoi et al. [18] proposed that reduced expression of αGlcNAc on MUC6 in premalignant lesions of the stomach, pancreas (IPMN), and esophagus (Barrett’s esophagus) may serve as an early event marking tumor progression. This suggests a potential common molecular alteration involved in the early stages of tumorigenesis in these organs.

Over the past two decades, several studies have described the higher frequency of EPMs in IPMN patients [5,8,12,13,22]. The strengths of our study include the following: the use of a population-wide cohort as a control group; assessing a heterogenous IPMN group (irrespective of surgery); and primary analyses with another control group of patients without a known IPMN undergoing EGDs. The novelty of the non-IPMN EGD group is that it overcomes the intrinsic bias of IPMN patients usually being under strict and long-term surveillance programs, making them more prone to detection of EPMs compared to the general population.

However, our study does have several limitations that should be acknowledged. Firstly, the cohort size was relatively small, which may have impacted the statistical power and generalizability of our findings. Secondly, there was missing data regarding the presence of pancreatic cysts in our control group, a limitation also noted in previous studies. Thirdly, the absence of clinical data, such as smoking history and other potential risk factors for gastrointestinal malignancies, might have influenced our results. Fourthly, the lack of histopathological specimens and genetic tests for IPMNs limited our ability to investigate shared molecular mechanisms underlying IPMNs and gastric cancer. Furthermore, we compared IPMN patients to individuals who underwent gastroscopy for other gastrointestinal reasons, which, although similar, may not be directly comparable. This might explain the comparable risk for gastric cancer observed in our study.

Regarding the study design, it is important to note that our research was cross-sectional in nature. Therefore, we were unable to determine the temporal relationship between the appearance of findings in the pancreas and the upper gastrointestinal tract. However, previous studies have shown that EPMs can develop years after surgical resection of IPMNs, suggesting that the temporal relationship between these processes may be irrelevant.

## 5. Conclusions

In conclusion, our study revealed a higher frequency of gastric cancer in IPMN patients compared to the general population. However, when compared to symptomatically matched individuals, the incidence of gastric cancer in IPMN patients did not show a significant difference. These findings highlight the need for further large-scale studies to better understand the role of upper endoscopies in routine surveillance for IPMN patients. Additional research is necessary to determine the optimal screening strategies and surveillance protocols for detecting and managing gastric cancer in individuals with IPMNs.

## Figures and Tables

**Figure 1 diagnostics-13-02127-f001:**
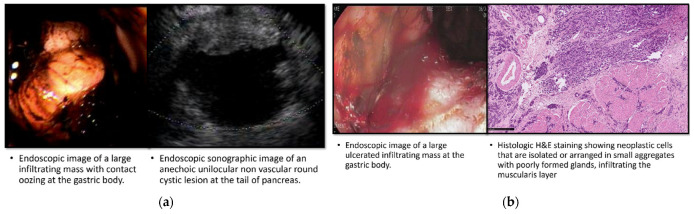
Endoscopic, sonographic, radiologic, and histologic imaging of IPMN patients diagnosed with gastric cancer in their first EGD. (**a**) Case 1; (**b**) Case 2; (**c**) Case 3; (**d**) Case 4.

**Table 1 diagnostics-13-02127-t001:** Clinical characteristic of patients with established IPMNs.

	Median	IQR
Age at EGD	70	61–76.5
Age at EUS	71	63–77
	n	Percentage
Male	123/340	36.1%
Personal h/o gastric cancer	3/340	0.9%
Family history of gastric cancer	1/340	0.3%
Personal h/o esophageal cancer	0	
Family history of esophageal cancer	1/340	0.3%
IPMN type (*n* = 340)		
SB-IPMN	314	92.3%
Mixed-type IPMN	15	4.4%
MD-IPMN	11	3.2%
IPMN with worrisome features	41	12%
IPMN with high-risk stigmata	10	2.9%
Size ≥ 30 mm	12	3.5%
Cyst size [10]	11	7–15
Cyst location (*n* = 340)		
Head	129	37.9%
Neck	77	22.6%
Body	139	40.8%
Tail	111	32.6%
Uncinate	65	19.1%
Multiple	117	34.4%
Worrisome features (*n* = 340)		
Mural nodule	12	3.5%
Cyst wall thickened	6	1.8%
Intracystic mass	7	2.1%
MPD dilatation	35	10.3%
Other associated pancreatic mass	11	3.2%
Cyst progression to pancreatic cancer	9	2.6%
EUS-FNA	146	42.9%
Cytology (*n* = 110)		
Acellular	25	22.7%
Benign cells	75	68.2%
Atypia	4	3.6%
Carcinoma	4	3.6%
Suspicious for carcinoma	2	1.8%
Clinical decision (*n* = 323)		
Imaging surveillance	305	94.4%
RF ablation	5	1.5%
Surgery consult	6	1.8%
Surgery	7	2.1%

EGD—esophagogastroduodenoscopy; EUS—endoscopic ultrasound; IPMN—intraductal papillary mucinous neoplasm; SB-IPMN—side-branch intraductal papillary mucinous neoplasm; MD-IPMN—main-duct intraductal papillary mucinous neoplasm; MPD—main pancreatic duct; EUS-FNA—endoscopic ultrasound fine needle aspiration; RF—radio frequency.

**Table 2 diagnostics-13-02127-t002:** EGD findings comparison between patients with established IPMNs vs. controls (gender, age at procedure, and year of endoscopy matched).

	IPMN Patients (*n* = 340)	Controls (*n* = 340)	*p* Value
Gastric cancer	4 (1.2%)	5 (1.5%)	>0.999
Esophageal cancer	1 (0.3%)	0	>0.999
Gastric LGD Adenoma	2 (0.6%)	2 (0.6%)	>0.999
Gastric HGD Adenoma	0	1 (0.3%)	>0.999
Gastric Hyperplastic polyp	21 (6.2%)	13 (3.8%)	0.16
Gastric NET	1 (0.3%)	1 (0.3%)	>0.999
Gastric GIST	0	0	
Barrett’s esophagus—No dysplasia	6 (1.8%)	10 (2.9%)	0.314
Barrett’s esophagus—Low grade dysplasia	1 (0.3%)	2 (0.6%)	>0.999
Barrett’s esophagus—High grade dysplasia	1 (0.3%)	0	>0.999
Duodenal NET	2 (0.6%)	0	0.5
Gastric Lymphoma	1 (0.3%)	0	>0.999
Ampullary tumor	5 (1.5%)	1 (0.3%)	0.217

IPMN—intraductal papillary mucinous neoplasm; LGD—low-grade dysplasia; HGD—high-grade dysplasia; NET—neuroendocrine tumor; GIST—gastrointestinal stroma tumor.

**Table 3 diagnostics-13-02127-t003:** Clinical and pathological characteristics of gastric cancer cases in the cohort.

Gastric Cancer Number	IPMN	Location	Pathology	Stage
1	Yes	Antrum	Mucin-producing adenocarcinoma	IIB
2	Yes	Body	Mucinous adenocarcinoma	IIIA
3	Yes	Body	Mucinous and signet ring cell adenocarcinoma	III
4	Yes	Fundus	Poorly differentiated carcinoma	IIA
5	No	Body	Poorly differentiated carcinoma	IV
6	No	Antrum	Poorly differentiated adenocarcinoma of stomach	IIIA
7	No	Body	Mucin-producing adenocarcinoma	IIB
8	No	Fundus	Adenocarcinoma of stomach	IIIA
9	No	Fundus	Mucin-producing adenocarcinoma	IIIB

IPMN—intraductal papillary mucinous neoplasm.

**Table 4 diagnostics-13-02127-t004:** Standardized incidence ratio calculation.

Age	Gender (Number)		Gender (Rate for 100,000)		Gender (Predicted Number)	
	0	1	0	1	0	1
<30	0	2	0	0.44	0.0000000	0.0000088
30.00–34.99	1	1	0	1.26	0.0000000	0.0000126
35.00–39.99	1	2	1.75	2.58	0.0000175	0.0000516
40.00–44.99	1	6	3.2	2.23	0.0000320	0.0001338
45.00–49.99	1	8	7.21	1.99	0.0000721	0.0001592
50.00–54.99	6	10	11.02	9.38	0.0006612	0.0009380
55.00–59.99	12	18	17.24	8.8	0.0020688	0.0015840
60.00–64.99	12	30	31.27	18.13	0.0037524	0.0054390
65.00–69.99	23	35	37.97	25.13	0.0087331	0.0087955
70.00–74.99	20	39	52.7	30.3	0.0105400	0.0118170
75.00+	46	67	88.97	47.32	0.0409262	0.0317044
Total					0.0668033	0.0606439
					Predicted=	0.127
					Observed=	4
					Obs/Pred	31.39
					95%CI	8.38–78.76

## Data Availability

The datasets used and analyzed during the current study are available from the corresponding author upon reasonable request.

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
