# Peer review of "Esophagogastroduodenal Findings in Patients with Intraductal Papillary Mucinous Neoplasms"

_diagnostics, 2023, doi:10.3390/diagnostics13122127_

Round 1

Reviewer 1 Report

In the present study authors assess the prevalence of extrapancreatic malignancies (EPMs) in a cohort of patients with intraductal papillary mucinous neoplasms (IPMNs). They analyzed 340 IPMNs compared with  age- and gender-matched controls. The overall incidence of esophagogastroduodenal conditions did not differ between IPMN pts and control. However when matched on national cancer registry data the incidence of gastric cancer in IPMN pts was higher than expected.

The manuscript is well written and although  the study have several limitations (small pts size...) they are all well acknowledged in the discussion section. 

study does have several limitations that should be acknowledged

Author Response

Thank you for your comment. As mentioned above, the limitations were acknowledged in the discussion section. see line 250-261 in the revised manuscript.

We do hope that this revised version will be found suitable for publication.

Yours sincerely,

Dr. Dana Zelnik Yovel

---------------------------------------------------------------------------------------------

Reviewer 2 Report

Some methodological details are missing: 

1) In Methods please provide more details concerning: 

- indications for performing gastroscopy in study group and control group; you have only written that indications were similar, which is not sufficient in research paper. 

- more details on matching control group; you have only written matching was based on age and relevant clinical characteristis; what were these "relevant clinical characteristics. Please provide details.  Gender was not taken into account?

2) In results section you have written that incidence of gastric cancer was higher  than expected in patients with IPMN compared to general population. And you gace the value of SIR. However, please provide detialsed calculations in the form of table or otherwise - to show how this result was obtained. 

Author Response

Attached is the R1 version of our research paper: " ESOPHAGOGASTRODUODENAL FINDINGS IN PATIENTS WITH INTRADUCTAL PAPILLARY MUCINOUS NEOPLASMS".

Revisions are highlighted with 'tracked changes' and were made according to the comments

We do hope that this revised version will be found suitable for publication.

Yours sincerely,

Dr. Dana Zelnik Yovel

Detailed answers to reviewer:

1. indications for performing gastroscopy in study group and control group; you have only written that indications were similar, which is not sufficient in research paper.

Response: The requested change was made in the attached revised manuscript. We added relevant information regarding the indication for EGD in each group ((dyspepsia, dysphagia, hematemesis/coffee ground, heartburn. Melena, nausea/vomiting, weight loss, diarrhea, anemia, occult blood, suspected celiac disease).

more details on matching control group; you have only written matching was based on age and relevant clinical characteristis; what were these "relevant clinical characteristics. Please provide details.  Gender was not taken into account?

Response: Thank you for your important comment. Gender was taken into account, and matching was based on gender as well as age and similar indications for EGD. A control group was established, consisting of patients who underwent an EGD for similar indications (dyspepsia, dysphagia, hematemesis/coffee ground, heartburn. Melena, nausea/vomiting, weight loss, diarrhea, anemia, occult blood, suspected celiac disease), but did not have a known diagnosis of IPMN. We clarified this in the revised manuscript.

2. In results section you have written that incidence of gastric cancer was higher than expected in patients with IPMN compared to general population. And you gave the value of SIR. However, please provide detailed calculations in the form of table or otherwise - to show how this result was obtained. 

Response: Thank you for this comment. A table demonstrating how the SIR was calculated was added to the revised manuscript.

Age

Gender (Number)

Gender (Rate for 100,000)

Gender (Predicted Number)

0

1

0

1

0

1

<30

0

2

0

0.44

0.0000000

0.0000088

30.00 - 34.99

1

1

0

1.26

0.0000000

0.0000126

35.00 - 39.99

1

2

1.75

2.58

0.0000175

0.0000516

40.00 - 44.99

1

6

3.2

2.23

0.0000320

0.0001338

45.00 - 49.99

1

8

7.21

1.99

0.0000721

0.0001592

50.00 - 54.99

6

10

11.02

9.38

0.0006612

0.0009380

55.00 - 59.99

12

18

17.24

8.8

0.0020688

0.0015840

60.00 - 64.99

12

30

31.27

18.13

0.0037524

0.0054390

65.00 - 69.99

23

35

37.97

25.13

0.0087331

0.0087955

70.00 - 74.99

20

39

52.7

30.3

0.0105400

0.0118170

75.00+

46

67

88.97

47.32

0.0409262

0.0317044

Total

0.0668033

0.0606439

Predicted=

0.127

Observed=

4

Obs/Pred

31.39

95%CI

8.38-78.76